# In Vivo Targeted Metabolomic Profiling of Prostanit, a Novel Anti-PAD NO-Donating Alprostadil-Based Drug

**DOI:** 10.3390/molecules25245896

**Published:** 2020-12-13

**Authors:** Ksenia M. Shestakova, Natalia E. Moskaleva, Natalia V. Mesonzhnik, Alexey V. Kukharenko, Igor V. Serkov, Igor I. Lyubimov, Elena V. Fomina-Ageeva, Vladimir V. Bezuglov, Mikhail G. Akimov, Svetlana A. Appolonova

**Affiliations:** 1Laboratory of Pharmacokinetics and Metabolomic Analysis, Institute of Translational Medicine and Biotechnology, I.M. Sechenov First Moscow State Medical University, 2-4 Bolshaya Pirogovskaya St., 119991 Moscow, Russia; ksenia.shestakova@labworks.ru (K.M.S.); nemoskaleva@gmail.com (N.E.M.); natalia.mesonzhnik@labworks.ru (N.V.M.); doctoralexey@inbox.ru (A.V.K.); svetlana.appolonova@labworks.ru (S.A.A.); 2Institute of Physiologically Active Compounds RAS, Severniy pr., 1, 142432 Chernogolovka, Russia; serkoviv@mail.ru; 3Territory of Skolkovo Innovation Center, LLC “Gurus BioPharm”, Bolshoy Boulevard, 42 Building 1, 143026 Moscow, Russia; lyubimov@gurus.bio; 4Shemyakin-Ovchinnikov Institute of Bioorganic Chemistry RAS, St. Miklukho-Maklaya, 16/10, 117997 Moscow, Russia; evfa56@gmail.com (E.V.F.-A.); vvbez@ibch.ru (V.V.B.)

**Keywords:** peripheral arterial disease, Prostanit pharmacokinetics, 1,3-dinitroglycerol, nitric oxide generation, Prostanit metabolomic response

## Abstract

Prostanit is a novel drug developed for the treatment of peripheral arterial diseases. It consists of a prostaglandin E_1_ (PGE_1_) moiety with two nitric oxide (NO) donor fragments, which provide a combined vasodilation effect on smooth muscles and vascular spastic reaction. Prostanit pharmacokinetics, however, remains poorly investigated. Thus, the object of this study was to investigate the pharmacokinetics of Prostanit-related and -affected metabolites in rabbit plasma using the liquid chromatography-mass spectrometry (LC-MS) approach. Besides, NO generation from Prostanit in isolated rat aorta and human smooth muscle cells was studied using the Griess method. In plasma, Prostanit was rapidly metabolized to 1,3-dinitroglycerol (1,3-DNG), PGE_1_, and 13,14-dihydro-15-keto-PGE_1_. Simultaneously, the constant growth of amino acid (proline, 4-hydroxyproline, alanine, phenylalanine, etc.), steroid (androsterone and corticosterone), and purine (adenosine, adenosine-5 monophosphate, and guanosine) levels was observed. Glycine, aspartate, cortisol, and testosterone levels were decreased. Ex vivo Prostanit induced both NO synthase-dependent and -independent NO generation. The observed pharmacokinetic properties suggested some novel beneficial activities (i.e., effect prolongation and anti-inflammation). These properties may provide a basis for future research of the effectiveness and safety of Prostanit, as well as for its characterization from a clinical perspective.

## 1. Introduction

Peripheral arterial disease (PAD), also known as obliterating endarteritis, is an inflammatory disorder mainly affecting arteries of the lower limb (2–3% of the total population) [1]. The late stages of the disease may result in acute limb ischemia, most likely associated with decreased life quality and limb amputation. The prostaglandin E_1_-based drug Alprostadil and its analogs pose a competitive alternative for limb amputation [2]. The main effect of prostaglandin E_1_ (PGE_1_) is antiplatelet activity and direct vasodilatation with subsequent changes in blood flow [3,4,5]. The predominant smooth muscle relaxation mechanism induced by PGE_1_ is the activation of G-protein coupled receptors with a consequent increase of intracellular cAMP synthesis followed by a lowering of intracellular calcium [6]. An increase of nitric oxide (NO) concentration and endothelial and neuronal, but not inducible, NO synthase content was also described [7]. Thus, Alprostadil improves blood flow both in ischemic tissues during severe obliterating disorders of peripheral arteries and in normal tissues. It leads to an improvement of tissue oxygen delivery and, at the same time, decreases lactate formation [8]. However, PGE_1_ utilization is also linked to several side effects; some of them are connected with severe ischemic limb diseases and limit the usage of the drug [9]. In addition, PGE_1_ does not address endothelial dysfunction, which occurs during the severe stages of vascular disorders. Therefore, despite a wide range of PGE_1_-based drugs presented on the pharmaceutical market [10,11], the development of more safe and potent multitarget analogs of Alprostadil is still an actual problem.

Nitric oxide (NO) is an important signaling molecule that mediates diverse processes in the cardiovascular system. It is synthesized from the amino acid l-arginine by an isoenzyme family known as NO synthases (NOS). The vasodilatory activity of NO is caused by its interaction with guanylyl cyclase followed by the elevation of cyclic guanosine monophosphate (cGMP) that activates protein kinases, resulting in relaxation of smooth muscle [12]. NO is also involved in the regulation of vascular inflammation and angiogenesis and provides antiplatelet and cytoprotective effects during ischemic injury [13,14,15,16,17].

The late stages of PAD are characterized by decreased ability of the vascular endothelium to produce vasodilation. This is mainly caused by impaired synthesis of endothelium-derived vasodilators, particularly NO and prostacyclin (PGI_2_) [18,19]. Therefore, given the violation of metabolic pathways of NO formation occurring at the late stage of endothelial dysfunction, the development of drugs with NO donors to compensate for these pathogenetic processes is a promising strategy for PAD treatment [20].

PGE_1_ dinitroglycerol ester, or Prostanit, was first described as a potential nitric oxide (NO)-donating PGE_1_ derivative with vasodilatory and antiplatelet properties [21,22]. This vasodilation agent has already been patented in Russia [23]. Unlike PGE_1_, Prostanit provides NO generation in smooth muscle cells both by means of NO synthase activation and due to the presence of NO-generating fragments. This action could compensate or enhance the vascular endothelium ability in NO synthesis and gives an opportunity for the application of Prostanit in the treatment and prevention of PAD. In addition, after Prostanit hydrolysis, the resultant PGE_1_ may have a synergistic effect with NO [24,25]. However, to optimize its application as a therapeutic agent, an understanding of its pharmacokinetics and metabolism is needed [26]. Drug-induced dynamic metabolomic alterations can also be useful for the identification of new health-promoting effects of a drug.

Generally, metabolomics has been successfully applied in many fields of medicine, including pharmaceutical studies [27,28]. At the same time, in studies of endogenous compounds in biological matrices, liquid chromatography-mass spectrometry (LC-MS) approaches are the most common [29].

In the present study, the pharmacokinetic properties of Prostanit were assessed in vivo in rabbits after a single intravenous injection of the drug at a dose of 40 µg/kg. It was found that Prostanit underwent rapid hydrolysis with consequent formation of 1,3-dinitroglycerol (1,3-DNG), PGE_1_, and 13,14-dihydro-15-keto-PGE_1_. Then, targeted LC-MS/MS analysis was applied to study the main pharmacologically significant metabolites of Prostanit (which are mostly endogenous compounds) and to identify the metabolic pathways affected by Prostanit. At the same time, the constant growth of certain amino acid levels (proline, 4-hydroxyproline, alanine, phenylalanine, etc.), steroids (androsterone and corticosterone), and purines (adenosine, adenosine-5 monophosphate, and guanosine) was observed. The levels of glycine, aspartate, cortisol, and testosterone were decreased. The obtained results pointed to the existence of both additional beneficial mechanisms of Prostanit action in the PAD setting, as well as several possible adversarial effects.

## 2. Results

### 2.1. Identification and Pharmacokinetics of Prostanit Metabolites

Prostanit contains in its structure multifunctional groups that may characterize metabolic transformations of the drug in vivo. There are two possible pathways of Prostanit metabolism after injection. The first one is associated with the hydrolysis of the glycerol ester bond and the consequent formation of endogenous PGE_1_ and 1,3-dinitroglycerol (1,3-DNG). PGE_1_ itself may be further metabolized by the action of 15-hydroxydehydrogenase after the pulmonary cycle, resulting in 15-keto metabolites. The second proposed pathway starts with nitric oxide liberation from the Prostanit molecule with the formation of the 2-glycerol ester of PGE_1_. Therefore, plasma levels of these metabolites, together with the source drug, were monitored in rabbits from vehicle control and Prostanit-treated groups.

We did not detect any quantity above the detection limit of Prostanit or PGE_1_ glycerol ester after the injection of this drug (data not shown). Thus, we excluded the second proposed pathway of Prostanit metabolism under the study conditions.

1,3-DNG appeared in plasma immediately upon Prostanit injection with a maximum concentration of 15 ng/mL at the 2nd minute (Figure 1a). The slightly elevated level of 1,3-DNG in the untreated group at the 6th minute was statistically non-significant and might have resulted from sample preparation errors.

The growth of PGE_1_ concentration in the treated group was found only at the first two time-points—2 and 4 min after the drug administration, with the maximum concentration at 4 min after Prostanit administration (Figure 1b). Up to the 30th minute, the PGE_1_ level remained stably elevated in the treated group with a concentration of 5 ng/mL (Figure 1b).

The maximum concentration of the secondary PGE_1_ metabolite 13,14-dihydro-15-keto-PGE_1_ was registered immediately after Prostanit administration at the second time-point (2 min, Figure 1c). The decrease of 13,14-dihydro-15-keto-PGE_1_ to the basal level was registered in 1 h after Prostanit administration; at this time point, no significant difference between treated and vehicle control groups was found.

Table 1 summarizes the main pharmacokinetic parameters of the identified Prostanit metabolites after a single intravenous injection of the drug.

### 2.2. NO Generation from Prostanit in Rat Isolated Aorta

Although Prostanit was initially characterized as a vasodilator, and its vasodilatory properties were shown to be dependent on the NO donor moiety [22], NO generation from the drug was never demonstrated directly. To measure NO generation (both from the drug and induced by the drug), we incubated Prostanit with rat isolated aorta and measured NO formation using the Griess reaction.

Prostanit addition to the aorta fragments led to substantial NO production (Figure 2). The NO concentration increase induced by Prostanit remained constant from 30 to 90 min incubation time. The incubation time (up to 90 min) did not allow for enzyme expression induction, and so the produced NO should be attributed either to the NO donor groups of the molecule or to NO synthase activation by the molecule.

### 2.3. NO Generation from Prostanit in Differentiated Human Smooth Muscle Cells

To further elucidate whether Prostanit can serve as an NO generator and activate NO synthase, a series of experiments with differentiated human smooth muscle cells was performed. The experiments included the NO synthase inhibitor L-N^G^-Nitro arginine methyl ester (L-NAME) to check the participation of the enzyme in the observed generation and PGE_1_ and 1,3-DNG to check their ability to influence NO synthase activity and NO synthase-independent NO generation, accordingly (Figure 3).

Prostanit induced rapid NO accumulation in the incubation medium. After 30 min of incubation, NO accumulation was partially inhibited by the NO synthase inhibitor. 1,3-DNG induced similar NO production, which was not sensitive to the NO synthase inhibitor. Almost no generation of NO was observed for PGE_1_.

### 2.4. Targeted Metabolite Changes Induced by Prostanit Treatment

During untargeted and targeted investigation of metabolic changes after injection of the structurally similar drug Nitroproston, a dinitroglycerol derivative of prostaglandin E_2_, Shestakova et al. showed that intravenous administration of Nitroproston mainly affected steroid, amino acid, purine, and pyrimidine metabolism [30]. Therefore, in this work, we measured a similar set of endogenous metabolites, including 21 steroids, 23 amino acids, 31 purines, and pyrimidines.

Plasma metabolites that increased significantly (*p* < 0.05) after treatment with Prostanit included steroids (corticosterone and androsterone), amino acids (alanine, proline, 4-hydroxy proline, isoleucine, phenylalanine, γ-aminobutyric acid, and aminoisobutyric acid), and purines (adenosine, adenosine 5-monophosphate, and guanosine). At the same time, steroids (testosterone and cortisol) and amino acids (glycine and aspartate) were significantly decreased (Figure 4 and Figure 5). The complete list of the metabolites assayed is provided in Appendix A.

Targeted metabolite profiles showed clear discrimination between the vehicle control and the treated groups over time (Figure 4, Table 2). Moreover, the metabolites, which were increased or decreased after Prostanit treatment, presented a consistent rise even after the second time point, demonstrating the prolonged effects of Prostanit administration.

### 2.5. Network Visualization and Analysis of Biochemical Pathways

To identify the biochemical pathways influenced by Prostanit treatment, we performed an enrichment analysis and network visualization on the targeted metabolites profiles using the MetaboAnalyst software (Figure 5 and Figure 6).

Prostanit administration induced changes in 26 metabolic pathways. Figure 6 represents the most significant (*p* < 0.05) metabolic pathways that were changed after drug treatment; the most modulated were purine, glutathione, androsterone, and alanine metabolic cycles. Other substantially affected pathways included glutamate, arginine, proline, and selenoamino acid metabolism, as well as the urea cycle, ammonia recycling, beta-alanine metabolism, and methionine metabolism. Most of the pathways were connected to a single network (Figure 6).

## 3. Discussion

In this work, a study of the pharmacokinetic properties of the drug Prostanit was performed to gather data for the optimization of its use as a potential anti-PAD agent. Three experimental systems were used: rabbits, rat isolated aorta, and human smooth muscle cell culture. In rabbits, although the drug was not detectable in the bloodstream 2 min after the injection, a constantly elevated level of two its bioactive metabolites, PGE_1_ and 1,3-DNG, was observed for more than 30 min. In rat isolated aorta and human primary smooth muscle, Prostanit induced the expected NO generation. In addition, targeted metabolites in the bloodstream demonstrated changes, which pointed to inflammation and stress level reduction and indicated enhanced collagen degradation.

Based on Prostanit structure and literature review, we proposed a scheme of Prostanit metabolism (Figure 7).

We did not detect any quantity of Prostanit two minutes after the injection of this compound (data not shown), but all of the predicted metabolites were identified. The concentrations of the terminal-inactivated 15-keto derivative and of 1,3-DNG returned to the basal level on the 30th–40th minute, while PGE_1_ remained elevated till the 60th minute. This behavior contrasts with that described for PGE_1_ (Alprostadil), for which during an intravenous administration of up to 80 µg of the substance to human volunteers, only the level of 15-keto-13,14-dihydro-PGE_1_ was elevated [31]. Given the higher stability of the 15-keto derivatives compared with the source PGE_1_ [32,33], the observed pharmacokinetic properties pointed to an accumulation of Prostanit in tissues after injection, followed by slow liberation of the drug. Such behavior could be interpreted as activity prolongation, which is undoubtedly beneficial for PAD treatment.

We also did not detect any 2-glycerol ester of PGE_1_ even at the first time point after Prostanit injection. Prostaglandin glycerol esters are novel cyclooxygenase 2 (COX2) metabolites of endocannabinoid 2-arachidonoyl glycerol and putatively other closely related glycerol esters of dihomo-γ-linolenic acid or eicosapentaenoic acid with biological properties distinct from those of corresponding natural prostaglandins [34]. The absence of 2-glycerol ester of PGE_1_ in the metabolite profile led us to the conclusion that the mentioned metabolite is not involved in the biological effects of Prostanit, at least in rabbits.

Due to the unstable nature of NO, the production of this molecule from Prostanit was studied in two ex vivo systems (Figure 2 and Figure 3). Prostanit demonstrated two phases of NO liberation: a quick one, which was finished within 10 min of incubation with cells or aorta fragments, and a slow one, which was detectable only after 30 min of incubation.

The first phase was similar to that of 1,3-DNG and was not inhibited by the NO synthase inhibitor, and thus represents the metabolism of the NO donor groups. The sources of NO in the cell culture medium are quite limited. They are: (1) the activity of NO synthase; (2) side reactions of the substance with NO detection reagents; and (3) the production from an NO donor. In the first case, the enzymatic activity should be inhibited by L-NAME, as it is not selective for any particular NO synthase isoform, and it works in the case of extended incubation times. However, it does not affect the short-term incubation results, and so the NO synthase participation should be ruled out. The side reaction of Prostanit in the cell culture medium was tested (Appendix A) and may account only for 5% of the observed NO production. Therefore, the only remaining option is NO generation from the NO donor. However, we cannot fully exclude the participation of the intracellular NO donors in the observed process, and this question may be interesting for an in-depth understanding of the Prostanit action mechanism.

The second phase was inhibited by the NO synthase inhibitor and thus is dependent on the activity of this enzyme. These activities of Prostanit were experimentally described here for the first time. The second phase of Prostanit-induced NO generation was too fast for the expression change of NO synthase to occur, and thus should be attributed to the stimulation of the activity of this enzyme. PGE_1_ was inactive in this setting, and thus it could be speculated that the observed stimulation is Prostanit-specific. In its structure, this drug is similar to the known bioactive lipid 2-arachidonoyl glycerol, for which the stimulation of NO synthase via cannabinoid receptor 1 (CB1) receptor was described [35], and thus it could be hypothesized that Prostanit acts via a similar mechanism. In the PAD setting, the existence of two NO generation phases could result in some variation of the Prostanit activity depending on the NO generation system status of the patient, and this could be accounted for during dose calculation.

In addition to the direct Prostanit metabolites, several amino acids, steroids, and purines in the rabbits’ bloodstream responded to drug injection (Figure 4 and Figure 5, Table 2). Thus, the amino acid levels of alanine, proline and its metabolite 4-hydroxyproline, isoleucine, phenylalanine, γ-aminobutyric and aminoisobutyric acids, and carnosine were elevated, while aspartate and glycine levels were decreased. Among steroids, endogenous levels of corticosterone and androsterone were elevated, while testosterone and cortisol decreased. Finally, purines (adenosine, adenosine monophosphate, and guanosine) were increased after Prostanit treatment.

The response of alanine, carnosine, and isoleucine pointed to the overall reduction of inflammation and activation of the protective systems after Prostanit administration, and thus should represent an additional beneficial action mechanism of the drug. This conclusion is supported by the following reasoning. First, it was shown that pro-inflammatory cytokines increase alanine utilization [36], and thus an increased concentration of this amino acid could represent its reduced consumption due to reduced inflammation level. In its turn, it is known that L-glutamine with L-alanine or as a dipeptide is able to induce cytoprotective effects mediated by heat shock protein 70 (HSP70)-associated responses to muscle damage and inflammation [37]. Isoleucine is known for its protective ability having a beneficial contribution to the vascular wall [38]. Carnosine possesses a broad range of protective properties, including hypoglycemic, anti-inflammatory, and anti-oxidative effects [39].

The observed increase of phenylalanine/tyrosine ratio is somewhat controversial in the context of PAD. On the one hand, this effect was documented in different conditions associated with increased oxidative stress and inflammation and thus may point to the development of adverse effects. On the other hand, this increased ratio may suggest diminished activity of the phenylalanine hydroxylase (PAH) enzyme by oxidation as well as tetrahydrobiopterin (BH4) deficiency, an essential cofactor of PAH [36]. Since BH4 is also a cofactor of nitric oxide synthase (NOS) and can be depleted by oxidative stress and inflammation, this ratio may be indicative of NO synthase activation by the drug, and thus an additional supplementation of arginine or BH4 precursors together with Prostanit could be beneficial.

The observed 4-hydroxyproline, aspartate, and glycine level changes may be some form of Prostanit side effects. Thus, the growth in the concentration of 4-hydroxyproline most probably is indicative of excessive collagen hydrolysis [40], which is a known effect of PGE_1_ [41]. Aspartate is known to inhibit atherogenesis and fatty liver disease in cholesterol-fed rabbits [42]. Glycine treatment is known to decrease the levels of oxidative stress markers in rats and increase the concentrations of glutathione and γ-glutamylcysteine and the amount of γ-glutamylcysteine synthetase, a key enzyme of glutathione biosynthesis [43], and thus the reduction of the levels of these two amino acids could be undesirable in PAD.

Overall, the observed changes in the steroid levels indicated a reduced level of stress and possible anti-inflammatory action of Prostanit. Cortisol, as the main glucocorticosteroid in the area of fasciculate of the adrenal cortex, is known to be responsible for biochemical stress. The salivary testosterone level of men also showed a significant increase under exam stress [44], and thus the absence of their increase after Prostanit injection as compared to the vehicle control may be associated with reduced stress level. Finally, corticosterone was shown to be a predominantly anti-inflammatory steroid hormone [45]. All of these effects were described for prostaglandins. Thus, PGE_1_ significantly increases the basal secretion of corticosterone [46], and prostaglandins were shown to block testosterone production via a direct effect on the testicular interstitial cells [47]. In addition, the decreased levels of cortisol may be associated with an inhibition of steroidogenesis by NO [48]. The described results, being beneficial in the PAD setting, may be not fully representative, as the rabbit groups contained only male subjects, and at least testosterone response has a substantial correlation with the sex [44].

Finally, the observed increase in purine levels is also beneficial for PAD, as they are key intermediates in cAMP and cGMP pathways involved in NO signaling and smooth muscle relaxation. These changes may be a result of an increased cAMP and cGMP turnover during Prostanit signaling.

It is interesting to note that most of the observed changes in the targeted metabolites concentrations were linked within a single network of biochemical pathways (Figure 6). This could point to the existence of one or several control points, influenced by Prostanit, and poses an interesting problem for future research.

Overall, in addition to the confirmation of the NO donor activity of Prostanit, the current study revealed the potentially beneficial and adverse influence of the compound on targeted metabolites. These effects, however, were observed in healthy animals and thus have a limited significance for the disease state. In this view, a study in a PAD model, especially in humans, poses substantial interest. If the discovered adverse Prostanit effects manifest in a PAD setting, additional research concerning a combined therapy with other drugs to mitigate these effects looks promising.

## 4. Materials and Methods

### 4.1. Chemicals and Reagents

Prostanit (1′,3′-dinitroglycerol ester 11(*S*),15(*S*)-dihydroxy-9-keto-13*E*-prostanoic acid) and prostaglandin (PGE_1_, PGB_1_, and PGE_2_) and 1,3-dinitroglycerol (1,3-DNG) reference standards were kindly provided by the Laboratory of Oxylipins of the Shemyakin-Ovchinnikov Institute of Bioorganic Chemistry of the Russian Academy of Sciences. The reference standards of 15-keto-PGE_1_ and 13,14-dihydro-15-keto-PGE_1_ were purchased from Santa Cruz Biotechnology (Dallas, TX, USA). The HPLC purity levels of all reference standards were above 99.0%.

Metabolite standards of amino acids, purines, pyrimidines, steroids, and corticosteroids were supplied by Sigma-Aldrich (Steinheim, Germany) and Fluka (Buchs, Switzerland). The isotopically labeled standards of amino acids were provided by Sigma-Aldrich, and deuterated PGE_1_-d_4_ was purchased from Toronto Research Chemicals (Toronto, ON, Canada).

Ultrapure water was obtained from a Millipore Milli-Q water purification system (Millipore Corporation, Billerica, MA, USA). Diethyl ether (HPLC Plus grade), ethyl acetate (HPLC Plus grade), ethanol, methanol, and sodium sulfate were purchased from Sigma-Aldrich (Steinheim, Germany). LC-MS/MS ultrapure water was obtained from Biosolve BV (Valkenswaard, The Netherlands), acetonitrile (HPLC grade) from AppliChem (Panreac, Darmstadt, Germany), and formic acid from Sigma-Aldrich (Steinheim, Germany). Human aorta smooth muscle cells (HAOSMC) culture medium was from Cell Applications, San Diego, CA, USA.

### 4.2. Stock and Working Solutions

The stock solutions of Prostanit (1 mg/mL), PGE_2_, 1,3-DNG, PGE_1_, 15-keto-PGE_1_, and 13,14-dihydro-15-keto-PGE_1_ were prepared in 50% ethanol.

The stock solution of the internal standard (IS), PGE_1_-d_4_, was dissolved in 50% ethanol to a concentration of 1 μg/mL and was stored at −80 °C. This solution was diluted to the working solution at a concentration of 10 ng/mL and was used during the sample preparation. All stock solutions were stored in amber glass vials at 4–8 °C before use.

The stock solutions of the analytes were further diluted using the blank rabbit plasma to obtain calibration and quality control (QC) standards. The internal standards of PGE_1_ at a concentration of 1000 ng/mL were added to the incubated samples during the termination step.

The standard stock solutions (ng/mL) of amino acids, steroids, purines, and pyrimidines were prepared in Milli-Q water, acetone, MeOH, or EtOH. The internal standards used for all stock solutions were stored in LC-MS amber vials at −80 °C for optimum stability. The calibration solutions were made for each class of analytes.

### 4.3. Sample Collection

Twelve male Chinchilla rabbits (2.5 ± 0.18 kg), randomly divided into two groups, received an equal volume of the target drug (treatment group, *N* = 6) or saline (vehicle control group, *N* = 6). Rabbits were eight months of age and acclimatized for five days before the experiments. Prostanit was administered to the treatment group through a marginal ear vein at the dose of 40 µg/kg, while rabbits from the vehicle control group received the same volume of saline. The collection of whole blood samples was performed at 0, 2, 4, 6, 8, 12, 18, 24, 32, 40, 48, and 60 min after treatment. Following blood collection, samples were centrifuged at 3000 rpm for 10 min to obtain plasma that was stored at −70 °C until the analysis. All the experiments and care of the rabbits were performed in strict compliance with Guidelines for the Use of Laboratory Animals and authorized by the Animal Ethics Committee of the Sechenov First Moscow State Medical University (protocol # 31–20 from 11.02.2020).

### 4.4. Sample Preparation

The methods were based on the procedures published by Shestakova et al. [30]. Sample preparation for the determination of direct Prostanit metabolites and steroid profiling was performed using liquid–liquid extraction. In brief, 15 μL of the internal standard (10 ng/mL PGE_1_-d_4_ solution in 50% ethanol for prostaglandins and 15 ng/mL methyltestosterone for steroids, purines, and amino acids), 150 μL of water, and 0.1 g Na_2_SO_4_ were added to 150 μL of each sample. After that, extraction of the mixture was performed by subsequent addition of 1 mL of ethyl acetate and 1 mL of diethyl ester. After 10 min centrifugation at 4 °C and 3000 rpm, the supernatants were pooled and evaporated under nitrogen. The residues were reconstituted in 50 μL acetonitrile and transferred into vials for the LC-MS analysis.

The sample preparation for amino acid, purine, and pyrimidine profiles was performed using the dilute-and-shoot approach with protein precipitation as follows: 400 μL of acetonitrile was added to 100 μL of each plasma sample, then the samples were vortexed, and the supernatant was transferred into the LC-MS vials for the analysis.

### 4.5. Instrumental Analysis

The LC-MS analysis was conducted using the UPLC ACQUITY system connected to a Xevo TQ-S micro IVD (in vitro diagnostic device) system (Waters Corporation, Milford, MA, USA). Metabolite identification, optimization of LC-MS/MS conditions, and quantification were performed using reference standards of 15-keto-PGE_1_ and 13,14-dihydro-15-keto-PGE_1_, amino acids, steroids, purines and pyrimidines. The instrument settings, including multiple reactions monitoring (MRM) transitions and validation parameters for the pharmacokinetic study as well as for the analysis of steroid, amino acid, purine, and pyrimidine profiles, are presented in Appendix A.

### 4.6. Analytical Validation

The performed LC-MS method was validated in accordance with the International Council for Harmonization of Technical Requirements for Pharmaceuticals for Human Use (ICH) guidelines and laboratory standard operational procedures for the bioanalytical method validation. The calibration standards and QC samples were prepared every validation day. The peak area ratios of calibration standards were proportional to the concentration of analytes in each assay over the entire analytical ranges. The calibration curves were linear. A weighting factor of 1/x was applied to achieve the homogeneity of variance. The correlation coefficients (*r*^2^) for all analytes were above 0.98. The calculated concentrations of the calibration standards were within 15% of their nominal values. The lower limit of quantification (LLOQ) was presented as the lowest standard on the calibration curve that could be quantitatively determined with acceptable precision and accuracy. Precision and accuracy of the assay were determined by performing the analysis of replicate QC samples (*N* = 6) at three concentrations on the same day and three consecutive days. The data from these QC samples were examined by a one-way analysis of variance (ANOVA). The results demonstrated that the values were within the acceptable ranges. Analytical validation reported the accuracy for each analyte as <20% for the lowest limit of quantification and <15% for all other quality control (QC) levels. The analytical precision (also in %CV (coefficient of variation)) for each analyte was in the range of 15%. To test the applicability of the method to multiplate and multiday preparations, a serum pool underwent periodic repeat analyses during a run. The %CV values obtained for each analyte were <15%. To evaluate the matrix effect in the experiment, chromatographic peak areas of each analyte from the spike after extraction samples at low and high concentration levels were compared to the neat standards at the same concentrations. Percent nominal concentrations estimated were within the acceptable limits (86.7–101.5%) after evaluating six different lots of plasma. The same evaluation was performed on the IS and no significant peak area differences were observed. Thus, ion suppression or enhancement from plasma matrix was negligible for this method. Mean extraction recoveries were more than 94.6% for all analytes. Observed effects were consistent and reproducible. The method showed good consistency throughout the entire standard concentration ranges.

Stability studies were performed for stock and working solutions of all analytes. The results revealed that all analytes were stable in stock and working solutions for 24 h at room temperature of about 25 °C, for 60 h in the autosampler at 10 °C, and in refrigerator at 4–8 °C.

Full information concerning the validation procedure is presented in Appendix A.

### 4.7. Data Processing and Statistical Analysis

Data processing was performed using the MassLynx software (Waters Corp., Milford, MA, USA). The distribution of each metabolite was performed by the assessment of the areas under the curve in treated and vehicle control groups using the Shapiro–Wilk test with the consequent discrimination of those significantly different using the Student *t*-test or equivalent non-parametric Mann–Whitney U test (in accordance with their distribution). The main metabolic pathways involved were visualized using an enrichment analysis and network visualizations by means of the MetaboAnalyst software (www.metaboanalyst.ca, Montreal, QC, Canada).

### 4.8. Rat Aorta Isolation

All the experiments and care of the rats were performed in strict compliance with Guidelines for the Use of Laboratory Animals and authorized by the Animal Ethics Committee of Sechenov First Moscow State Medical University. Aorta extraction was performed at 4 °C, freshly for each experiment. Young male Wistar rats (body weight, 200 ± 10 g) were anesthetized using diethyl ether and subjected to cervical dislocation. Thereafter, the aorta was removed, cut first longitudinally and then in small rings, washed in ice-cold phosphate-buffered saline (PBS), and weighted.

### 4.9. NO Generation from Prostanit with Rat Aorta Fragments

During the study of NO generation from Prostanit, rat aorta fragments of ~10 mg weight were placed in 200 µL of PBS, pH 7.4, with or without 156 µM of the drug, and incubated for 90 min at 37 °C. Every 30 min, a 50 µL aliquot of the incubation medium was removed and subjected to NO determination. Each incubation was done in triplicate.

### 4.10. NO Quantification

NO concentration was determined using the modified Griess method [49]. In brief, to 50 µL of the incubation medium, 50 µL of a 1:1 mixture of 2% sulfanilamide in 5% H_3_PO_4_ and 0.04% *N*-1-naphtyletylenediamine in H_2_O was added and incubated for 10 min at room temperature in darkness. After that, the optical density of the reaction mixture was determined photometrically at λ_max_ = 540 nm using an Efos 9505 plate reader (MZ Sapfir, Moscow, Russia). The concentration curve was generated using a freshly prepared NaNO_2_ solution in PBS, with PBS as the blank control.

### 4.11. NO Generation by Human Smooth Muscle Cells

Differentiated human smooth muscle cells were treated with the substances for 10, 30, or 180 min. The substances were prepared as stock solutions in ethanol, dissolved in the culture medium, and added to the cells as 100 µL fresh medium with the substances to 100 µL of the old medium in the wells of 96-well plates. Some samples were additionally treated with 100 µM of L-NAME; the inhibitor was added 30 min before the substances. After incubation, the medium was collected and the content of NO in it was determined using the modified Griess method.

### 4.12. Cell Culture

Human adult aorta smooth muscle cells (Cell Applications, San Diego, CA, USA, cat. no. 354-05a) were maintained in the Human SMC Growth Medium (Cell Applications, San Diego, CA, USA). To induce cell differentiation, the cells were seeded in a 96-well plate at the density of 15,000 cells per well, and the medium was replaced with the Human SMC Differentiation Medium (Cell Applications, San Diego, CA, USA) for two weeks.

## 5. Conclusions

In summary, this paper argued that the novel anti-PAD drug Prostanit underwent rapid hydrolysis with a consequent appearance of its main bioconversion products—1,3-dinitroglycerol, PGE_1_, and 13,14-dihydro-15-keto-PGE_1_, the first two being pharmacological active. The metabolomic investigation clearly demonstrated that Prostanit administration provoked elevated formation of bioactive molecules that act in the same direction as PGE_1_ and 1,3-DNG, counteracting PAD pathology. The obtained pharmacometabolomic results allowed us to hypothesize that the main mechanism of action of Prostanit is an active vasodilation effect provided by the activation of both cAMP and cGMP pathways and nitric oxide liberation. Ideally, these findings should be replicated in a human study. Nevertheless, these results provide a good starting point for discussion and further research of the novel pharmaceutical Prostanit.

## Figures and Tables

**Figure 1 molecules-25-05896-f001:**
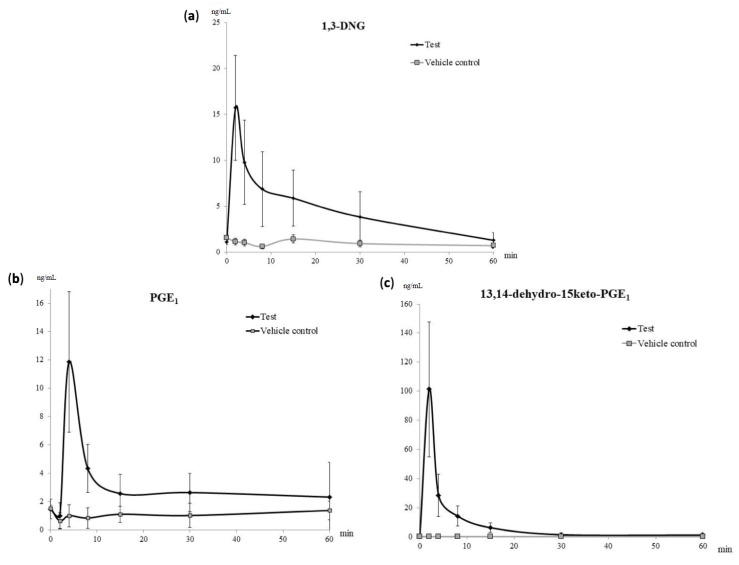
Pharmacokinetic curves of the main Prostanit metabolites found in the plasma of treated and vehicle control rabbits. (**a**) The pharmacokinetic curve of 1,3-dinitroglycerol (1,3-DNG); (**b**) pharmacokinetic curve of prostaglandin E_1_ (PGE_1_); (**c**) pharmacokinetic curve of 13,14-dihydro-15-keto-PGE_1_, *N* = 6 animals.

**Figure 2 molecules-25-05896-f002:**
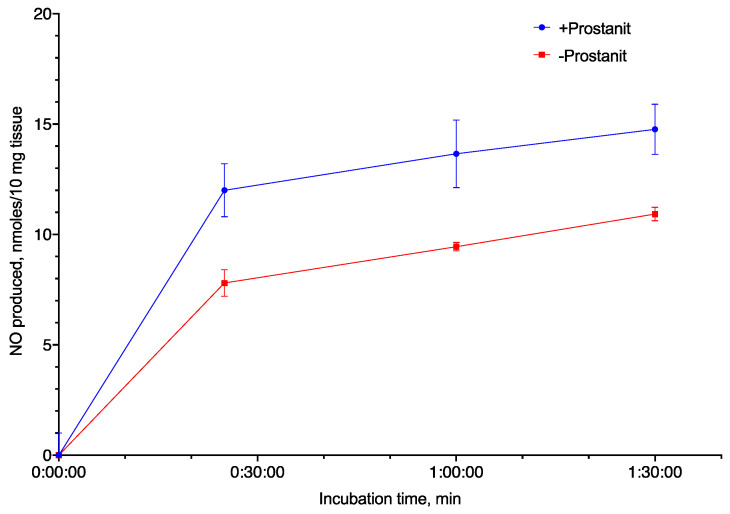
Nitric oxide (NO) generation from Prostanit by rat aorta fragments (10 mg). Prostanit concentration = 156 µM, NO quantification using the Griess method, mean ± SD (*N* = 3 experiments); the differences at all time points except 0 min are significant (ANOVA with Tukey’s post-test, *p* < 0.05).

**Figure 3 molecules-25-05896-f003:**
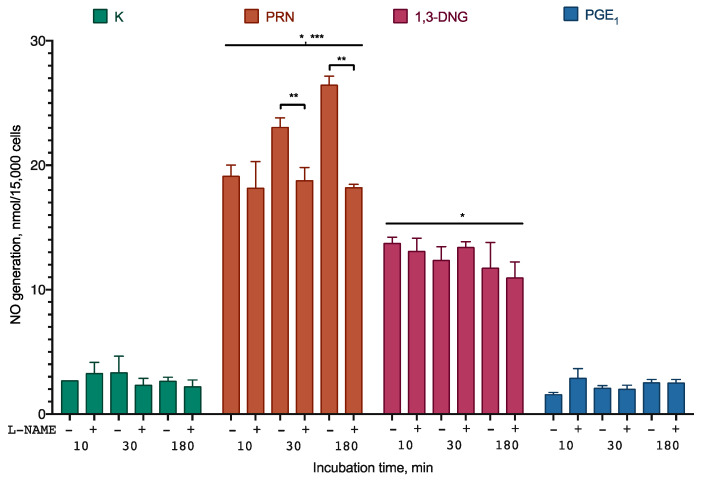
NO generation from Prostanit and its analogs by differentiated smooth muscle cells. The cells were treated with 35 µM of Prostanit or 90 µM PGE_1_ or 1,3-DNG with or without 100 µM of the NO synthase inhibitor L-NAME (L-N^G^-Nitro arginine methyl ester). PRN, Prostanit; 1,3-DNG, 1,3-dinitroglycerol; PGE_1_, prostaglandin E_1_. Griess assay data, mean ± SD; *, a statistically significant difference from the control (K); **, a statistically significant difference from the samples with L-NAME; ***, a statistically significant difference from 1,3-DNG; *p* < 0.05 in ANOVA with Tukey’s post-test.

**Figure 4 molecules-25-05896-f004:**
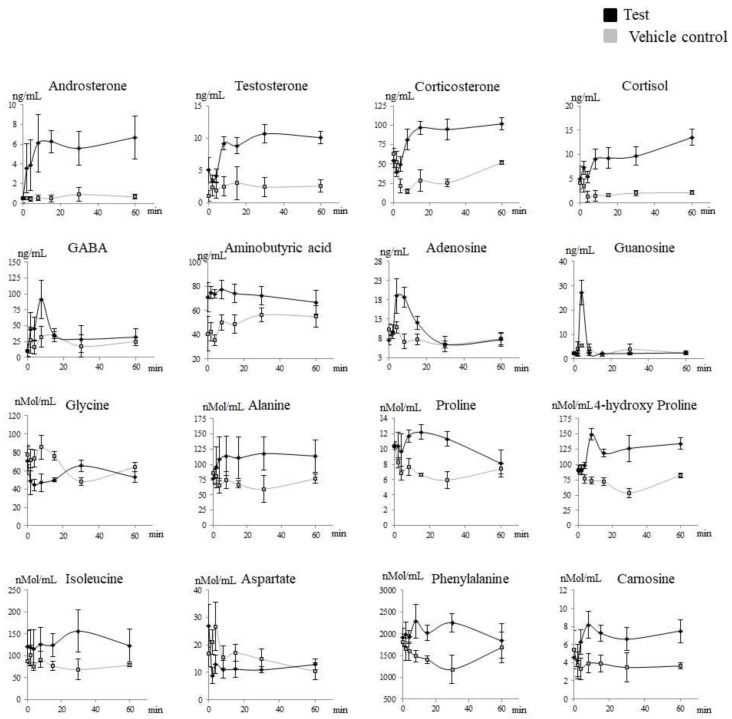
Targeted metabolites that significantly changed after Prostanit administration. Treated (black) versus vehicle control (gray) groups. X-axis, time after the drug administration. Y-axis, the content of the appropriate metabolite in rabbit plasma. The assessment of the areas under the curve in treated and vehicle control groups using the Shapiro–Wilk test with the consequent discrimination of those significantly different using the Student *t*-test or equivalent non-parametric Mann–Whitney U test (in accordance with their distribution).

**Figure 5 molecules-25-05896-f005:**
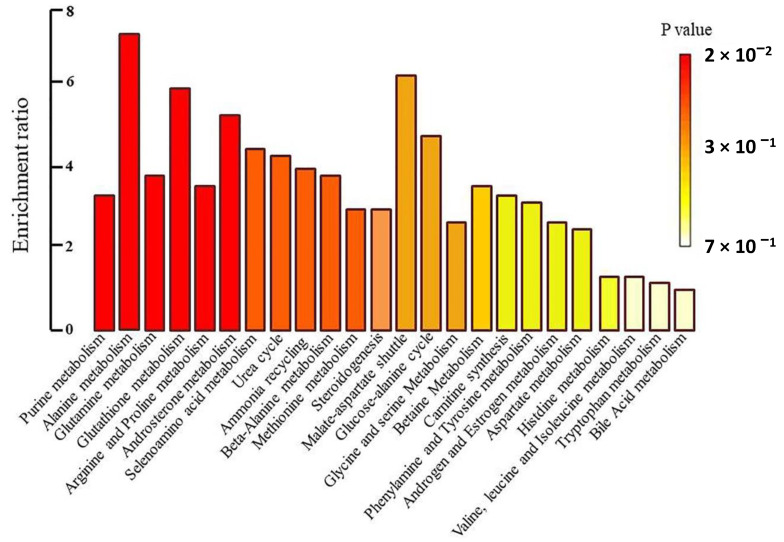
Metabolite set enrichment overview.

**Figure 6 molecules-25-05896-f006:**
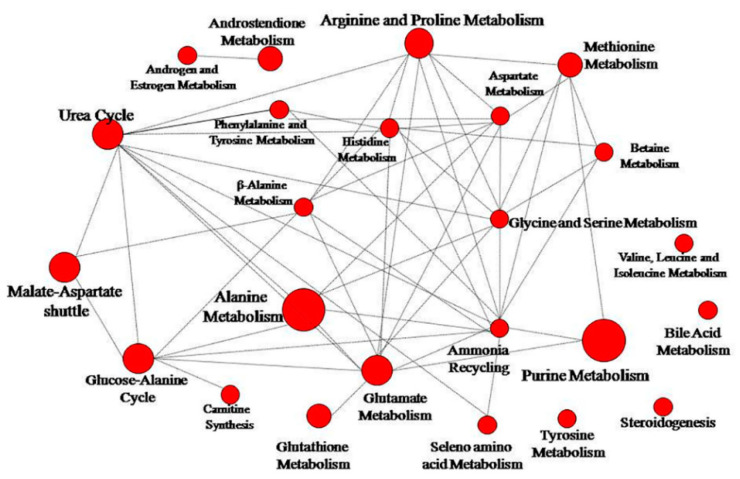
Network pathway visualization analysis. Higher size of the nodes reflects higher level of relevance for each pathway.

**Figure 7 molecules-25-05896-f007:**
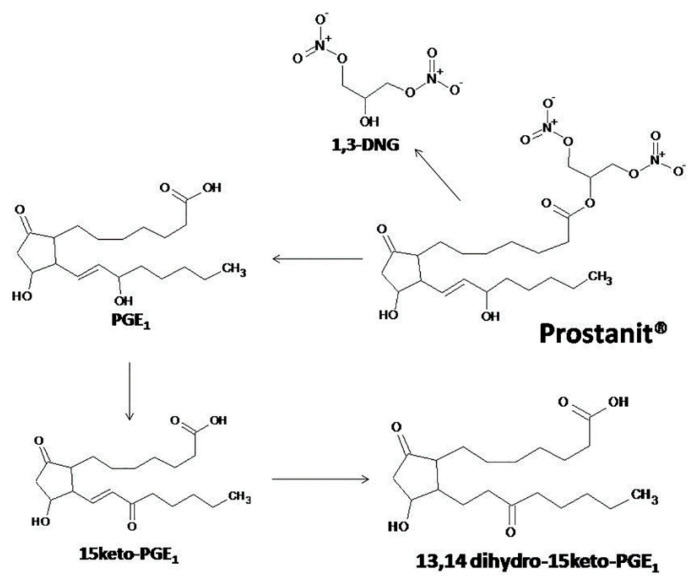
The proposed metabolism of Prostanit.

**Table 1 molecules-25-05896-t001:** Pharmacokinetic parameters of the identified Prostanit metabolites.

Name of the Metabolite	C_max_ (ng/mL)	T_1/2_ (min)
1,3-DNG	16.27 ± 5.8	16.87 ± 6.9
PGE_1_	10.8 ± 1.9	17.05 ± 2.7
13,14-dihydro-15-keto-PGE_1_	101.44 ± 41.1	11.42 ± 4.6

**Table 2 molecules-25-05896-t002:** List of the metabolites that were significantly altered after intravenous administration of Prostanit in rabbits. *p*-Values were calculated by comparison of the areas under the curve (AUC) of the analyzed metabolites in tested and vehicle control groups using parametric or non-parametric a/b tests (Student *t*-test and Mann–Whitney test, respectively).

Name of the Metabolite	Class	Calculated *p*-Value
Androsterone	Steroids	7.73 × 10^−7^
Testosterone	Steroids	4.15 × 10^−^^4^
Corticosterone	Steroids	1.08 × 10^−10^
Cortisol	Steroids	5.55 × 10^−11^
GABA	Amino acid	0.035
Aminoisobutyric acid	Amino acid	3.49 × 10^−6^
Adenosine	Purines	2.71 × 10^−^^4^
Guanosine	Purines	0.031
Glycine	Amino acid	4.67 × 10^−−^^4^
Alanine	Amino acid	0.0191
Proline	Amino acid	1.49 × 10^−5^
4-Hydroxyproline	Amino acid	7.44 × 10^−8^
Isoleucine	Amino acid	3.86 × 10^−^^3^
Aspartate	Amino acid	5.55 × 10^−^^11^
Phenylalanine	Amino acid	6.57 × 10^−^^4^
Carnosine	Amino acid	4.037 × 10^−7^

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
