# Peer review of "In Vivo Targeted Metabolomic Profiling of Prostanit, a Novel Anti-PAD NO-Donating Alprostadil-Based Drug"

_molecules, 2020, doi:10.3390/molecules25245896_

Round 1

Reviewer 1 Report

Comments for Authors:

------------------------------------------------------------------------------------------------------------------------

The word “besides” is distracting from the intended sentence.

48 Besides, an increase of NO concentration and endothelial and neuronal, but

49  not inducible NO synthase content was also described [7].

------------------------------------------------------------------------------------------------------------------------

The word bloodstream is inappropriate here, do the authors mean bloodflow? Content? Characteristics?

49…improves the

 50  bloodstream

------------------------------------------------------------------------------------------------------------------------

This sentence is awkward.  “…by A decreased vascular…”?

65 The late stages of PAD are characterized by decreased vascular endothelium ability to produce

66 vasodilation.

------------------------------------------------------------------------------------------------------------------------

Problem with conclusion of test:

2.2. NO generation from Prostanit® in rat isolated aorta

Figure 2 shows substantial NO production in the absence of Prostanit.  Therefore the conclusion that NO is directly produced from the drug is not conclusive. 

To measure NO generation from the drug, we incubated Prostanit® with rat isolated aorta and measured NO formation using the Griess reaction.

This sentence implies that NO comes from the drug and not the tissue affected by the drug.

Is this what the authors mean?

Needs a re-write

Figure 3 and the preceding paragraph present the information far better.

------------------------------------------------------------------------------------------------------------------------

Discussion

Missing control experiments showing different initial NO levels with different administered concentrations.  The conclusion that NO donor groups contribute solely to the first spike was not proven.

Experimental

The purity

313 levels of all reference standards were considerably above 99.0 %.

To what level? How was it determined and why the estimation if known for certain?

Author Response

Point 1. The word “besides” is distracting from the intended sentence.

48 Besides, an increase of NO concentration and endothelial and neuronal, but

49  not inducible NO synthase content was also described [7].

Response 2: The word “besides” was removed.

Point 2. The word bloodstream is inappropriate here, do the authors mean bloodflow? Content? Characteristics?

49…improves the

 50  bloodstream

Response 2: The word “bloodstream” was replaced with “bloodflow”

Point 3. This sentence is awkward.  “…by A decreased vascular…”? 

65 The late stages of PAD are characterized by decreased vascular endothelium ability to produce

66 vasodilation.

Response 3: The sentence word order was corrected. 

Point 4. Problem with conclusion of test: 

2.2. NO generation from Prostanit® in rat isolated aorta 

Figure 2 shows substantial NO production in the absence of Prostanit.  Therefore the conclusion that NO is directly produced from the drug is not conclusive. 

Response 4: The conclusion was modified to include enzyme activation by Prostanit

Point 5. To measure NO generation from the drug, we incubated Prostanit® with rat isolated aorta and measured NO formation using the Griess reaction.

This sentence implies that NO comes from the drug and not the tissue affected by the drug.

Is this what the authors mean?

Needs a re-write

Response 5: The sentence was re-written to include both the NO generation from the drug and induced by the drug. 

Point 6. Discussion

Missing control experiments showing different initial NO levels with different administered concentrations.  The conclusion that NO donor groups contribute solely to the first spike was not proven.

Response 6: The sources of NO in the cell culture medium are quite limited. They are: 1) the activity of NO synthase; 2) side reaction of the substance with the NO detection reagents; 3) the production from NO donor. In the first case, the enzymatic activity should be inhibited by L-NAME, as it is not selective for any particular NO synthase isoform, and it works in the case of extended incubation times. However, it does not affect the short-term incubation results, and so the enzyme participation should not be the case. The side reaction of Prostanit in the cell culture medium was tested (and added to the Supplementary data) and may account only for 5% of the observed NO production. Therefore, the only remaining option is NO generation from the NO donor. However, we cannot fully exclude the participation of the intracellular NO donors in the observed process, and this was added to the discussion section.

Point 7. Experimental 

The purity

313 levels of all reference standards were considerably above 99.0 %.

 To what level? How was it determined and why the estimation if known for certain?

Response 7: The structure of the standard samples that were synthesized in Shemyakin-Ovchinnikov Institute of Bioorganic Chemistry was confirmed by NMR and mass spectrometry. Purity was assessed by HPLC.

Reviewer 2 Report

Author performed PK, drug metabolism and metabolomic study of Prostanit in 3 biological systems. The studies are interesting and provide DMPK data for the drug along with metabolomic profiles. 

However, the author is not able to integrate DMPK data with metabolomic perturbations upon drug treatment using the Pharmacometabolomics approach. We suggest the author to adopt the pharmacometabolomics approach from previous publications particularly “Pharmaco-metabonomic phenotyping and personalized drug treatment Nature volume 440, pages1073–1077(2006)” and “An Integrative Approach for Identifying a Metabolic Phenotype Predictive of Individualized Pharmacokinetics of Tacrolimus Clinical Pharmacology & Therapeutics (2010) 87 4, 426–436.). This will integrate the whole study and provide a better conclusion from the data generated from PK, drug metabolism and metabolomic analysis. 

Also, several key analytical methods related to metabolomics analysis are missing which should be provided to ensure the quality of the data. 

  1. In abstract: what does this sentence mean “.....Prostanit 25 ® primary and targeted metabolites in rabbit plasma using the LC-MS approach….”?
  2. The use of ‘targeted metabolomic’ analysis terminology is confusing. Author should clarify if he means targeted drug metabolite anlaysis or targeted metabolome analysis?
  3. Line 86; describes PK analysis using targeted metabolomic approach, which is not correct terminology as per metabolomics and DMPK research conventions. 
  4. Results 2.1: what is the dose of  Prostanit?
  5. Figure 3: please describe the abbreviations used in the figure legends.
  6. Results 2.4: Author used during untargeted metabolomic profiling for targeted metabolite analysis which is not ideal or author is again confusing the terminologies and types of analyses performed.
  7. Figure 4: has important metabolite profiling information. However, redundant bar plots make it hard to grasp the profiling patterns. Author can use better visualization to show comprehensive effect of drug on the metabolite profiles. 
  8. Methods: The author should describe more about metabolomics LC-MS analysis particularly for metabolite identification, quality control and data analysis.

Author Response

Point 1. Author performed PK, drug metabolism and metabolomic study of Prostanit in 3 biological systems. The studies are interesting and provide DMPK data for the drug along with metabolomic profiles. 

However, the author is not able to integrate DMPK data with metabolomic perturbations upon drug treatment using the Pharmacometabolomics approach. We suggest the author to adopt the pharmacometabolomics approach from previous publications particularly “Pharmaco-metabonomic phenotyping and personalized drug treatment Nature volume 440, pages1073–1077(2006)” and “An Integrative Approach for Identifying a Metabolic Phenotype Predictive of Individualized Pharmacokinetics of Tacrolimus Clinical Pharmacology & Therapeutics (2010) 87 4, 426–436.). This will integrate the whole study and provide a better conclusion from the data generated from PK, drug metabolism and metabolomic analysis. 

Response 1:  Thank you for the valuable additions. Pharmacometabolic approaches described in the proposed articles suggest investigating the possibility of predicting pharmacokinetic properties from metabolomic data. But the purpose of the work was the primary study of metabolism of Prostanit. After administration, Prostanite is metabolized to form endogenous compounds, which are rapidly involved in metabolic pathways. In our work, we tried to identify pharmacologically significant metabolites of Prostanite, as well as metabolic pathways that it affects. This approach was also used in the study of a similar drug Nitroprostone. (Shestakova, K., Brito, A., Mesonzhnik, N.V. et al. Rabbit plasma metabolomic analysis of Nitroproston®: a multi target natural prostaglandin based-drug. Metabolomics 14, 112 (2018). https://doi.org/10.1007/s11306-018-1413-1).

Point 2. Also, several key analytical methods related to metabolomics analysis are missing which should be provided to ensure the quality of the data. 

All the necessary analytical methods were added to the Experimental part as well as to the Supplementary tables. In abstract: what does this sentence mean “.....Prostanit 25 ® primary and targeted metabolites in rabbit plasma using the LC-MS approach….”?

Response 2: Thank you for your comment. This sentence is unclear. We changed it to “.....Prostanit 25 ® related and affected metabolites in rabbit plasma using the LC-MS approach….”

Point 3. The use of ‘targeted metabolomic’ analysis terminology is confusing. Author should clarify if he means targeted drug metabolite anlaysis or targeted metabolome analysis?

Response 3: The use of the term "targeted metabolomic profiling" in this study is explained by the fact that the metabolites of Prostanit® are endogenous substances. Also, in this work, we have studied the changes in other endogenous metabolites that can can be found in the body.The use of the term is explained in lines 88-91: “In this work targeted metabolomic profiling was applied both to study the main pharmacologically significant metabolites of Prostanit® (which are mostly endogenous compounds), and to identify the metabolic pathways mostly affected by Prostanit®”.

Point 4. Line 86; describes PK analysis using targeted metabolomic approach, which is not correct terminology as per metabolomics and DMPK research conventions. 

Response 4: Thank you for important correction. This part was changed (lines 85-90): “In the present study, the pharmacokinetic properties of Prostanit® were assessed in vivo in rabbits after a single intravenous injection of the drug in the dose of 40 mg/kg. It was found that Prostanit® undergoes rapid hydrolysis with consequent formation of 1,3-DNG, PGE1, and 13,14-dihydro-15-keto-PGE1. Then targeted LC-MS/MS analysis  was applied both to study the main pharmacologically significant metabolites of Prostanit® (which are mostly endogenous compounds), and to identify the metabolic pathways affected by Prostanit®.

Point 5. Results 2.1: what is the dose of  Prostanit?

Response 5: Thank you for your comment. As noted in Materials and methods section «Prostanit® was administered to the treatment group through a marginal ear vein in a dose of 40 µg/kg, while rabbits from the vehicle control group received the same volume of saline». An error in the Introduction section was corrected.

Point 6. Figure 3: please describe the abbreviations used in the figure legends.

Response 6: The abbreviations were added to the figure legend.

Point 7. Results 2.4: Author used during untargeted metabolomic profiling for targeted metabolite analysis which is not ideal or author is again confusing the terminologies and types of analyses performed.

Response 7: Thank you for important corrections.  This chapter provides a link to the Nitroprostone study.  This article presents the results of both untarget metabolome profiling, which was used to identify changing metabolites, and the results of targeted analysis. We changed the wording to be more precise: “During  untargeted and targeted investigation of metabolic changes after injection of the structurally similar drug Nitroproston®, a dinitroglycerol derivative of prostaglandin E2, Shestakova et al. showed that intravenous administration of Nitroproston® mainly affected steroid, amino acid, purine, and pyrimidine metabolism [30]

Point 8. Figure 4: has important metabolite profiling information. However, redundant bar plots make it hard to grasp the profiling patterns. Author can use better visualization to show comprehensive effect of drug on the metabolite profiles. 

Response 8: Thank you very much. We have changed the drawing to be more informative, please see line 175

Point 9. Methods: The author should describe more about metabolomics LC-MS analysis particularly for metabolite identification, quality control and data analysis.

Response 9: Thank you for important corrections. In this work, we used the analytical techniques we published earlier in the article  (Shestakova, K., Brito, A., Mesonzhnik, N.V. et al. Rabbit plasma metabolomic analysis of Nitroproston®: a multi target natural prostaglandin based-drug. Metabolomics 14, 112 (2018). https://doi.org/10.1007/s11306-018-1413-1).     Link to this article have been added to the section     4.4 (please, see line 356).      Information on metabolite identification and method validation has also been added to Sections 4.5 and 4.6. Condition of LC-MS analysis described in Supplementary materials.